# The Effect of Chinese Agarwood Essential Oil with Cyclodextrin Inclusion against PCPA-Induced Insomnia Rats

**DOI:** 10.3390/molecules28020635

**Published:** 2023-01-07

**Authors:** Yinfang Lai, Liping Hua, Jiali Yang, Juewen Xu, Junduo Chen, Shuangshuang Zhang, Shunyao Zhu, Jingjing Li, Senlin Shi

**Affiliations:** School of Pharmaceutical Sciences, Zhejiang Chinese Medical University, Hangzhou 311400, China

**Keywords:** agarwood, supercritical fluid extraction, β-cyclodextrin, nose-to-brain delivery, insomnia

## Abstract

Objective: To study the extraction process of agarwood active ingredients (AA) and investigate the safety and effectiveness of AA in the treatment of insomnia rats by nasal administration. Method: A β-cyclodextrin (β-CD) inclusion compound (a-β-CD) was prepared from agarwood essential oil (AEO), and the preparation process was optimized and characterized. The safety of AA in nasal mucosa was evaluated through *Bufo gargarizans* maxillary mucosa and rat nasal mucosa models. Insomnia animal models were replicated by injecting *p*-chlorophenylalanine (PCPA), conducting behavioral tests, and detecting the expression levels of monoamine neurotransmitters (NE and 5-HT) and amino acids (GABA/Glu) in the rat hypothalamus. Results: The optimum inclusion process conditions of β-CD were as follows: the feeding ratio was 0.35:1.40 (g:g), the inclusion temperature was 45 °C, the inclusion time was 2 h, and the ICY% and IEO% were 53.78 ± 2.33% and 62.51 ± 3.21%, respectively. The inclusion ratio, temperature, and time are the three factors that have significant effects on the ICY% and IEO% of a-β-CD. AA presented little damage to the nasal mucosa. AA increased the sleep rate, shortened the sleep latency, and prolonged the sleep time of the rats. The behavioral test results showed that AA could ameliorate depression in insomnia rats to a certain extent. The effect on the expression of monoamine neurotransmitters and amino acids in the hypothalamus of rats showed that AA could significantly reduce NE levels and increase the 5-HT level and GABA/Glu ratio in the hypothalamus of insomnia rats. Conclusion: The preparation of a-β-CD from AEO can reduce its irritation, improve its stability, increase its curative effect, and facilitate its storage and transport. AA have certain therapeutic effects on insomnia. The mechanism of their effect on rat sleep may involve regulating the expression levels of monoamine neurotransmitters and amino acids in the hypothalamus.

## 1. Introduction

Agarwood is the wood of *Aquilaria sinensis* (Lour.) Gilg, which has sedative, anti- inflammatory, antibacterial, antitumor, and other pharmacological effects [1,2,3]. Insomnia, a sleep disorder caused by sleep interruption or sleep time reduction, is often accompanied by different degrees of depression and anxiety symptoms, which will increase the risk of coronary heart disease and Alzheimer’s disease [4,5,6,7]. In serious cases, this disorder may cause a series of social problems. Clinical treatments for insomnia mainly include drug therapy (benzodiazepines, antihistamines, tricyclics, melatonin receptor stimulants, etc.) and non-drug therapy (psychotherapy, cognitive behavioral therapy, transcranial magnetic stimulation, etc.) [8,9,10]. Due to the complexity of insomnia and the low penetration rate of non-drug treatment, there is no convenient and minimal side-effect treatment for insomnia. Therefore, it is very important to find new drugs to regulate and treat insomnia. Agarwood has sedative and hypnotic effects. This plant has a history of thousands of years in treating neurological diseases, as recorded in ancient books such as *Compendium of Materia Medica*. The active ingredients of agarwood are mainly essential oils, including sesquiterpenes, chromones, aromatics, and fatty acid compounds [11,12]. AEO has sedative and hypnotic effects. Agarotetrol is a chromone component, and its content can reflect the quality of agarwood [13]. Thus, the present work studies AEO and its extract powder containing agarotetrol (EA) [14]. The traditional alternative method of agarwood is incense replacement therapy. At present, this treatment is mainly administered orally or in the form of incense. However, there are some problems, such as difficulty in controlling the dosage and slow efficacy. Nasal administration typically delivers drugs through absorption from the nasal mucosa to the blood circulation and other tissues to exert therapeutic effects [15,16,17,18]. Agarwood often treats brain diseases by inhalation, so we hypothesized that the active substance of agarwood may bypass the blood–brain barrier and enter the brain directly by nasal administration, achieving brain targeting in the route of administration, thus being conducive to the treatment of insomnia. Consequently, we paid more attention to the route of nasal administration [19,20].

This experiment optimized the inclusion process of AEO by response surface methodology to prepare a-β-CD and encapsulated the active ingredients of agarwood by β-CD to prevent volatile components from undergoing oxidative deterioration or volatilization loss, to reduce irritation, and to improve stability [21]. To provide safety information for insomnia pharmacodynamic experiments, the toad palate and rat nasal mucosa model were used to investigate the effects of agarwood’s medicinal components on mucosal integrity and mucociliary movement and to comprehensively evaluate nasal mucociliary toxicity [22]. *p*-Chlorophenylalanine (PCPA) [23] was used to establish the insomnia rat model to investigate the anti-insomnia effect of the agarwood active ingredients (AA), and it was combined with AEO/a-β-CD and EP [24]. The evaluation methods of anti-insomnia effects included the experiments of pentobarbital-sodium-induced sleep, the tail suspension test (TST) [25], the open field test (OFT), and the elevated plus-maze test (EPMT) [26], as well as the expression levels of monoamine neurotransmitters and amino acids [27].The mucosa toxicity evaluation was conducted using toad palate and rat nasal mucosa.

## 2. Results and Discussion

### 2.1. Chemical Composition of Agarwood

The GC–MS results are shown in Table 1 and Figure 1a. AEO contained α-santalol, β-guaiene, corymbolone, agarospirol, stigmasterol, γ-sitosterol, and other substances, which had neuroprotective effects and had hypnosis, anti-oxidation, reflex, anti-sleep, and other effects. According to the results of HPLC, as shown in Figure 1b–d, this method could be used to determine the content of agarotetraol in EA accurately and stably. It was found that EA contained a high content of agarotetraol.

### 2.2. Process Investigation and Characterization of a-β-CD

#### 2.2.1. Determination of AEO in a-β-CD by UV Analysis

At 227 nm, AEO and a-β-CD have the maximum absorption, and β-CD had almost no detected absorption peak, indicating that β-CD would not affect the absorption of AEO and a-β-CD. The method establishment results showed that the linear regression equation was y = 0.0196x + 0.0285 with R^2^ = 0.9993, indicating a good linear relationship in the range of 4.384–43.840 μg/mL. In the precision test, stability test, and repeatability test, the average ± SD of absorbance was 0.53 ± 0.00, 0.66 ± 0.01, and 0.32 ± 0.00, respectively; the RSD was 0.08%, 1.65%, and 0.52%; and the average recovery ± SD for AEO was 100.93 ± 2.84%, with an RSD of 2.81% in the sample recovery test. All of the values met the requirements, and the method was feasible.

#### 2.2.2. Single Factor Investigation of the β-CD Inclusion Process

The results of varying the feed ratio are shown in Table 2. When the feed ratio was 0.50:1.40, the ICY and OIC% were higher. When the feed ratio was 0.25:1.40, the IEO was higher. In the washing process, it was found that when the feed ratio was 0.50:1.40, more AEO was not included, so the feed ratio was selected to be 0.25:1.40. The influence of the inclusion temperature was investigated, when the inclusion temperature was 45 °C, the ICY, OIC, and IEO were all higher than those of the temperatures which were 40 °C and 50 °C, so the temperature was selected to be 45 °C. The influence of the inclusion time was also investigated. When the inclusion time was 3 h, the ICY and IEO were higher, but when this time was 2 h, the OIC was higher, so the inclusion time was set to 3 h. The process parameters were as follows: a feeding ratio of 0.25:1.40, an inclusion temperature of 45 °C, and an inclusion time of 3 h.

#### 2.2.3. Box–Behnken Design (BBD) Response Surface Methodology for Optimization of the Cyclodextrin Inclusion Process of AEO [28]

Based on the multivariable regression analysis model, the quadratic polynomial equations of ICY (%) and IEO (%) were measured as follows: $\mathrm{ICY}\%\left(X\right)=15.74*A+1.28*B-1.27*C+2.17*A*B+1.81*A*C-0.23*B*C-15.35{*A}^{2}-11.67{*B}^{2}-6.56{*C}^{2}$; $\mathrm{IEO}\%\left(Y\right)=2.50*A+2.41*B+0.58*C-1.27*A*B-0.47*A*C+3.00*B*C-16.31{*A}^{2}-16.87{*B}^{2}-7.94{*C}^{2}$.

The contour and response surface maps were drawn with Design-Expert 8.0.6 software, and the analysis of variance and data processing are shown in Table 3 and Table 4. The interaction between the various factors is shown in Figure 2. Figure 2a–c showed that when A was constant, X was basically constant with the increase in C, and the other three factors of A, B, and C first increased and then decreased the influence on X. Figure 2d–f shows that the three factors of ABC have an interactive effect on Y, the influence of these three factors on Y first increases and then decreases; the effects were in the order A—feeding ratio > B—inclusion temperature > C—inclusion time. The final determination of the optimized process was the feeding ratio of 0.35:1.40 (g:g), inclusion temperature of 45 °C, inclusion time of 2 h, and the theoretical prediction value of ICY%, 61.25%, and of IEO%, 65.53%. According to the optimized process, the actual ICY% and IEO% were 53.78 ± 2.33 (%) and 62.51 ± 3.21 (%), respectively. Therefore, these three factors possessed a remarkable influence on the ICY% and IEO% of a-β-CD.

#### 2.2.4. Appearance of a-β-CD

The results are shown in Figure 3. β-CD and a-β-CD were white, fine, powdery, and tasteless; the physical mixture was golden, irregular, and granular, with the smell of AEO.

#### 2.2.5. FT-IR Analysis

According to the infrared spectrum in Figure 4b, the stretching vibration of free OH and associated OH group in the β-CD molecule causes strong characteristic absorption at 3700–3100 cm^−1^; the stretching vibration of “-CH2–O–CH2-” in β-CD causes a strong absorption peak at 1050 cm^−1^. There are 2-(2-phenylethyl) chromones and sesquiterpenes in AEO, which have characteristic absorption peaks at 1750–600 cm^−1^. The infrared absorption of the physical mixture is different from that of β-CD in the range of 1160–1600 cm^−1^. In a-β-CD, the absorption frequency caused by the “−C=O” stretching vibration shifts to a low wavenumber (1330 cm^−1^). This shift may be due to the formation of a-β-CD by hydrogen bonding between AEO and β-CD and the change in electron cloud density, resulting in the shift in the characteristic absorption peak frequency of the groups in a-β-CD; alternatively, it may be that the included guest is wrapped within the hydrophobic cavity of β-CD, and the original characteristic peak is covered by the characteristic peak of β-CD. The infrared absorption of a-β-CD is different from that of the physical mixture mainly at 1650 cm^−1^ and 1350 cm^−1^, indicating the formation of a-β-CD [29].

#### 2.2.6. XRD Analysis

XRD is often used to study the crystal structure and lattice information of the host-guest complex. As shown in Figure 4c, the main diffraction peaks of β-CD are 10.6°, 18.8°, 21.3°, 22.8°, 44.7°, 65°, and 78.2° (2θ); the main diffraction peaks of the physical mixture are 10.6°, 12.5°, 18.8°, 22.8°, 27.2°, 31.9°, 34.8°, and 51.5° (2θ); the main diffraction peaks of a-β-CD are 10.6°, 12.5°, 18.8°, 22.8°, 27.2°, 31.9°, 34.8°, 51.5°, and 65° (2θ); and they all present crystalline structures. The main diffraction peaks of a-β-CD are different from those of the physical mixture and β-CD, indicating the formation of a-β-CD [30].

#### 2.2.7. SEM Analysis

It could be seen from Figure 4d that β-CD was lumpy and compact; the physical mixture was irregular and loose in texture; and a-β-CD was a small flake crystal with a clear edge, and its particle size and shape were completely different from those of β-CD and the physical mixture, which indicated the formation of inclusion compounds. Identified using FT-IR, XRD, and SEM, the a-β-CD was successfully prepared [31].

### 2.3. Safety Evaluation of Nasal Mucosa

The mucosa toxicity evaluation was conducted using toad palate and rat nasal mucosa.

#### 2.3.1. The Effect on the Cilia of Bufo Gargarizans Maxillary Mucosa In Vitro/In Vivo

Through analysis with SPSS software—the results are shown in Table 5—it could be seen that in the in vitro and in vivo experiments, the PVD of the positive control group was significantly different from that of the normal group and the agarwood groups (*p* < 0.001); what is more, there was no significant difference between the normal group and the agarwood groups. The results showed that there was almost no damage to toad maxilla mucosa, which provided a safety basis for the further study of nasal mucosa drug delivery preparations.

#### 2.3.2. The Observation Results of Ciliary Movement in the Maxillary Mucosa in Bufo Gargarizans

Macroscopic observation of the positive control group showed that hyperemia and red blood filaments were present on the surface of the maxillary mucosa after administration, and the mucosa was damaged, as shown in Figure 5. There was no significant change in the other groups. In the sodium deoxycholate group, serious cilia detachment and tissue edema occurred; in the other groups, there were fewer detached cilia, and the mucosa was intact.

### 2.4. Effect on Rat Nasal Mucosal Cilia

#### 2.4.1. H&E Staining Observation

As shown Figure 6, the epithelium of the normal group and the agarwood groups were complete, without inflammatory cell infiltration; the epithelium of the positive control group was incomplete, with more inflammatory cell infiltration, focal hemorrhage, and local necrosis.

#### 2.4.2. SEM Observation

As seen in Figure 6, the SEM observation showed that on the 8th day, a large number of cilia in the positive control group fell off, and the basal layer was seriously damaged; the cilia in the low-dose group and a-β-CD group were arranged more orderly, and there was no significant difference compared to the normal group; the cilia in the middle-dose and high-dose groups were arranged disorderly, and some cilia detached. On the 15th day, the cilia of the positive control group did not recover, and new cilia were formed in the middle-dose and high-dose groups; on the 22nd day, the positive control group did not recover, which indicated that sodium deoxycholate irreversibly destroyed the nasal mucosa tissue. In the middle-dose and high-dose group, the cilia movement of the nasal mucosa was damaged to some extent. After 22 days, the mucosa of *Bufo gargarizans* cilia recovered their activity, which proved that the ciliotoxicity was reversible. In general, these results confirmed that AA had good safety in the nasal mucosa of rats and could be used to treat insomnia through nasal administration.

### 2.5. The Effect of AA on Sleep in Rats with PCPA-Induced Insomnia

In this study, we used nasal mucosa administration to avoid the first pass effect of liver metabolism and to improve the efficacy by achieving brain targeting of the drug delivery pathway. The experimental results demonstrated that AA had a sedative–hypnotic effect.

#### Weight Change and Independent Activities

The weight of the animals was recorded every day and was found to decrease significantly after modeling (Figure 7a). The results showed that the differences between the model group and the agarwood groups were statistically significant in (Figure 7b), indicating that AA could reduce the number of autonomic activities in insomnia animals in a dose-dependent manner, and the a-β-CD group was the best, which is consistent with previous reports [32].

### 2.6. Open Field Test

In the open field experiment, the distance of entering the central region (Figure 7c) was in the order of the middle-dose group > α-β-CD group > high-dose group; the times of entering the central region (Figure 7d) was in the order of the α-β-CD group > high-dose group > middle-dose group; there was no statistical significance between the model group and the low-dose group. Compared to the model group, the low-, middle- and high-dose groups showed an increase in the total distance (Figure 7e) of exercise and a reduction in the number of feces (Figure 7f) in a dose-dependent manner, and the α-β-CD treatment was better than the high-dose treatment. The diazepam group had the lowest number of activities and lowest total distance in the open field test. The reason may be that diazepam has a strong sedative and hypnotic effect, which can lead to a decrease in the number of autonomous activities carried out by the animals. This appears contradictory to the exploratory nature of rats. The less time spent in ambulation may be interpreted as easily reaching the comfortable state of sleepiness and hence, forgoing the need to investigate the novel environment [1].

### 2.7. Tail Suspension Test

The results showed the immobility time of the model group was prolonged after PCPA injection; the diazepam and agarwood groups showed a reduced immobility time, which was significantly different from that in the model group (Figure 8a). The results implied that the animals in the agarwood groups exhibited a reduced immobility time, indicating that AA could alleviate the depression of insomnia rats, and the curative effect was in the following order: high-dose group > medium-dose group > low-dose group; and a-β-CD group > medium-dose group.

### 2.8. Elevated Cross Maze Test

The results of the elevated cross maze test are shown in Figure 8. The agarwood group increased the number of rats entering the open arm (Figure 8b) and the ratio of the time between entering the open arms and the total arms (Figure 8c). There were significant differences (*p* < 0.05) in the number of times entering the closed arm (Figure 8d) and the total distance into the maze (Figure 8e) between the agarwood groups and the model group, suggesting that AA can alleviate depression in insomnia rats. The diazepam group had lower activity times and lower total distances in the test of the elevated cross maze, which was consistent with the results of the open field test.

### 2.9. Sleep Induction in Rats by Drugs Combined with Pentobarbital Sodium

Compared with the model group, the agarwood groups showed an improved sleep rate, shortened sleep latency, and prolonged sleep time. The results of the sleep rate (Figure 9a) and latency time (Figure 9b) were in the order of medium dose > high dose ≥ low dose; the dose relationship showed a peak shape, and the effect in the a-β-CD group was better than that in the high-dose group in Figure 9. The effects of the low-dose, middle-dose and high-dose treatments on sleep time (Figure 9c) were dose dependent. These results showed that AA and pentobarbital had a synergistic effect, which reduced the sleep latency and prolonged the sleep time, implying the hypnotic effect of AA [33,34].

### 2.10. The Effects on Monoamine Neurotransmitters and Amino Acids in the Hypothalamus of Rats

It is of great significance to study the effect of AA on the NE, 5-HT level, and GABA/Glu ratio in the hypothalamus, and to evaluate AA treatment in insomnia. Neurotransmitters plays a key role in the transmission of nerve signals between cells. Sedative and hypnotic drugs often exert their pharmacological effects by regulating the expression level of neurotransmitters [33]. The NE level in the hypothalamus of insomnia rats was higher than that in the normal group and decreased after administration in Figure 9d. 5-HT plays an important role in sleep–wake regulation [34]. The results showed the expression of 5-HT in the hypothalamus of animals was increased in Figure 9e. The middle dose had the best effect, followed by the high-dose and a-β-CD treatments. The ratio of GABA/Glu in the insomnia rats was higher than in the normal rats in Figure 9f. The expression levels of GABA and Glu in the agarwood groups were increased and showed a dose-dependent relationship. It has been reported that in rats with neurological diseases, normal night sleep might be the key to maintaining brain health [27]. Improving sleep quality may help reduce the risk of neurological diseases.

## 3. Conclusions

In this paper, AEO and EA were used in combination to improve the utilization rate of agarwood. A-β-CD was prepared from AEO by cyclodextrin inclusion technology, and the preparation process was investigated and optimized. A-β-CD can reduce the irritation of AEO, improve stability, and facilitate storage and transport. The results of pharmacodynamics experiments showed that nasal administration of AA has a certain therapeutic effect on sleep disorder model rats and is relatively safe. The effect of AA on the sleep mechanism of insomnia rats may be realized by regulating the expression levels of monoamine neurotransmitters (NE, 5-HT) and amino acids (GABA, Glu). In this paper, the pharmacological mechanism of agarwood to improve sleep is relatively shallow, the follow-up can be an in-depth study. The a-β-CD treatment was better than the high-dose treatment at the same dose. The reason may be that a-β-CD causes less irritation to the nasal mucosa and distributes more evenly in the nasal cavity. After inclusion, the water-soluble macromolecular substance β-CD is more easily absorbed into the brain by the nasal mucosa than the lipid-soluble substance AEO. The above experiments showed that AA had a sedative and hypnotic effect and has a definite therapeutic effect on insomnia. When administered through nasal mucosa, the medicinal substances of agarwood can bypass the blood–brain barrier and enter the brain directly; to realize the brain targeting of drug administration is beneficial to the treatment of insomnia and other brain diseases, and to provide a safe and effective treatment for insomnia. In conclusion, AA can be used as a reference for the development of anti-insomnia products, is a prospective treatment in insomnia measures, and has important significance.

## 4. Methods and Materials

### 4.1. Materials

The agarwood (Hainan, China), AEO, EA, and a-β-CD were homemade. The AEO was extracted through supercritical fluid extraction technology, which is a superior method for extracting a higher yield and better quality of essential oil in a shorter time, compared to hydrodistillation [35]. The EA was obtained by ethanol reflux extraction. The processes for AEO and EA were optimized through a single factor investigation and BBD response surface methodology (the specific parameters are under patent protection and are temporarily undisclosed.) The AEO and EA were mixed according to the extraction ratio to prepare the low-, medium- and high-dose groups; a-β-CD and EA were mixed according to the ratio to form the a-β-CD group. After preparation of each group of drugs, they were stored at 4 °C for further use. All of the operations were in accordance with international health guidelines.

The 5-HT, NE, GABA, and Glu ELISA kits for the of rats were all purchased from Jiangsu Enzyme and Immunobiology Technology Co., Ltd. (Jiangsu, China).

### 4.2. Chemical Composition of Agarwood

#### 4.2.1. Chemical Composition of the AEO

GC–MS technology (Agilent 7890B-5977A gas chromatography-single quadrupole mass spectrometer, Agilent Company, Santa Clara, CA, USA) was used to analyze the components of the AEO extracted by supercritical fluid technology. 10 μL of the AEO sample was dissolved with 10 mL of absolute ethanol, the sample was filtered with a 0.22 μm syringe filter, and then it was transferred to the GC–MS sample bottle. A HP-5MS capillary column (0.25 mm × 30 m, 0.25 μm) was used. In the splitless mode, the high purity helium was used as a carrier gas at a constant flow rate of 1 mL/min, and 5 μL of the AEO sample was injected into the system. The injector port temperature was kept at 255 °C, the mass transfer line temperature was maintained at 300 °C, the oven temperature was programmed to 50 °C (hold for 2 min), increased to 120 °C at a rate of 10 °C/min, then 1 °C/min to 140 °C, 0.5 °C/min to 170 °C, and 5 °C/min to 300 °C, ending with a 5 min hold. According to the mass spectral library research (NIST), the compounds were identified by their retention time, and the mass fragmentation diagrams and composition analysis summary tables of all of the components were obtained.

#### 4.2.2. The Determination of Agarotetraol in EA by HPLC

The content of agarotetraol in EA was determined by HPLC (Agilent 1200, Agilent Company, Santa Clara, CA, USA) in Chinese Pharmacopoeia (2020 Edition). Acetonitrile was used as mobile phase A, 0.1% formic acid was used as mobile phase B, the column temperature was 30 °C, and the detection wavelength was 252 nm.

### 4.3. Process Investigation of a-β-CD

#### 4.3.1. Preparation of a-β-CD and Determination of Its Oil Content

To prepare the a-β-CD inclusion complex, the magnetic stirring method was used. AEO was accurately weighed, dissolved in 70% ethanol aqueous solution, diluted, and prepared for use. The β-CD was from Source Leaf Biology (Shanghai, China) and the a-β-CD was weighed, dissolved in 70% ethanol aqueous solution by ultrasonication, and stored until further use. A 2450-UV spectrophotometer (Shimadzu, Kyoto, Japan) was used to perform full wavelength scanning in the range of 190–800 nm. According to relevant regulations, a standard curve was established for the precision test, stability test, repeatability test, and the sample recovery test evaluations [36].

#### 4.3.2. Single Factor Investigation of the Cyclodextrin Inclusion Process

Approximately 2 g of β-CD was weighed, 40 mL of distilled water was added, the mixture was heated to dissolve it to prepare a saturated aqueous solution, and 0.25 mL of AEO was added dropwise under stirring. The magnetic stirrer speed was 300 r/min, the temperature was 40 °C, and the inclusion time was 2 h. After the mixture was cooled to room temperature, it was refrigerated at 4 °C for 24 h, filtered by a Brinell funnel, washed with distilled water and absolute ethanol three times in turn, and dried in a 45 °C oven for 4 h to obtain the inclusion compound [21,36,37].

Three factors were selected as the main investigation aspects: a. the feeding ratios (the ratio of essential oil to β-CD, g:g): 0.13:1.40, 0.25:1.40, and 0.50:1.40; b. the inclusion temperatures: 40 °C, 45 °C, and 50 °C; and c. the inclusion times: 1 h, 2 h, and 3 h. The inclusion compound yield (ICY, %), the oil content of the inclusion compound (OIC, %), and the inclusion rate of the essential oil (IEO, %) were taken as the indexes to determine the optimal values as the basis for the subsequent design of the experiment.

#### 4.3.3. Box–Behnken Design (BBD) Response Surface Methodology for Optimization of the Cyclodextrin Inclusion Process of AEO

According to the Box–Behnken central composite experimental design principle and single factor experiment, three investigation factors were selected: the feeding ratio (A), inclusion temperature (B), and inclusion time (C). Three levels were chosen for each factor, and X (ICY, %) and Y (IEO, %) scores were taken as the response values.

ICY% = oil content in a-β-CD/total weight of a-β-CD ∗ 100%

OIC% = the actual weight of the a-β-CD/(β-CD addition amount + AEO addition amount) ∗ 100%

IEO% = actual oil content in the a-β-CD/AEO addition amount ∗ 100%

### 4.4. Characterization of a-β-CD

#### 4.4.1. Fourier Transform Infrared (FT-IR) Spectroscopy Analysis

The appropriate amount of β-CD, the physical mixture, and a-β-CD were analyzed by a Nicolet IS50 infrared spectrometer (Thermo Fisher Scientific, Waltham, MA, USA). The wavenumber range was 400–4000 cm^−1^ [38].

#### 4.4.2. X-ray Diffraction (XRD) Analysis

An X-ray diffractometer (Shimadzu, Kyoto, Japan) was used to analyze β-CD, the physical mixture, and a-β-CD. The parameters were as follows: Cu target (40 kV, 40 mA), step scan: 0.02°/step, scan range 10°–80°, and scanning speed: 2°/min [22].

#### 4.4.3. Scanning Electron Microscopy (SEM) Analysis

After spraying gold on an appropriate amount of β-CD, the physical mixture, and a-β-CD, the samples were fixed on the sample table with conductive double-sided adhesive, and their morphological characteristics were observed under vacuum by an SU8010 field emission scanning electron microscope (Hitachi, Japan) [39].

### 4.5. Safety Evaluation on the Nasal Mucosa in Bufo Gargarizans

In the safety evaluation test of AA to the nasal mucosa, 72 *Bufo gargarizans* toads weighing 30 to 40 g were used, and for the in vitro experiment and in vivo experiment, 36 of them were randomly divided into groups, and the groups were as follows: a. the normal group (normal saline); b. the positive control group (1% sodium deoxycholate (Sigma-Aldrich, Saint Louis, MO, USA); c. the low-dose group: a mixture of AEO (12.43 mg/kg) and EA (3.63 mg/kg), converted into a crude drug amount (0.29 g/kg); d. the middle-dose group: a mixture of AEO (24.85 mg/kg) and EA (7.28 mg/kg), converted into a crude drug amount (0.58 g/kg); e. the high-dose group: a mixture of AEO (49.70 mg/kg) and EA (14.55 mg/kg), converted into a crude drug amount (1.16 g/kg); and f. A-β-CD: a mixture of a-β-CD (79.52 mg/kg) and EA (14.55 mg/kg), converted into a crude drug amount (1.16 g/kg). This was carried out by referring to the scope of application of agarwood in *Chinese Pharmacopoeia* (2020 Edition) and the preliminary experimental results.

#### 4.5.1. Evaluation of Mucociliary Movement of Toad Maxillary Mucosa In Vitro

The spinal cords of the toads were damaged with a probe and fixed on a frog plate in the supine position. The mouth was pulled with hemostatic forceps to prevent the toad from swallowing the drugs and to expose its maxillary mucosa [40]. Two pieces of maxillary mucosa were isolated from each toad, approximately 3 mm × 3 mm. The surface of the maxillary mucosa was spread on the glass slide with the surface facing upward. Normal saline was added to the surface of one mucosa for self-control, agarwood was added to the other mucosa, and the movement of maxillary mucosa was observed under an inverted fluorescence microscope (Thermo Electron Corporation, Waltham, MA, USA). The specimens were observed with a microscope at every fixed time. The time from the beginning of observation to the cessation of mucociliary movement, which is called the persistent vibration duration (PVD), was recorded.

PVD % = duration of maxillary mucosa continuous movement in the agarwood groups/duration of maxillary mucosa continuous movement in the normal group × 100%

#### 4.5.2. Evaluation of Mucociliary Movement of Toad Maxillary Mucosa In Vivo

The toad was fixed on the frog board on its back, and 0.5 mL of liquid medicine was dripped on the maxillary mucosa to completely immerse the mucosa. After the drug had fully contacted the maxilla for 30 min, the mucosal surface was cleaned. After the mucosa was stripped (3 mm × 3 mm), the follow-up steps were the same as in “Section 2.4.1”. The continuous movement time of the maxillary mucosa and PVD (%) were recorded.

### 4.6. The Effect on the Cilia of Nasal Mucosa in Rats

Seventy-two rats were randomly divided into 6 groups with 12 rats in each group. The experimental group of the nasal mucosa safety of the rats was the same as that of the maxillary mucosa toxicity test of Bufo gargarizans in vivo. A microinjector was inserted into the nasal cavity of the rats for administration. Each rat was given 50 μL of solution each day for 7 days. The drug treatment was stopped on the 8th day. Six rats in each group were anesthetized on the 8th day, and then, three rats in each group were anesthetized on the 15th and 22nd days. The mucosa on both sides of nasal septum were taken to observe the morphology and integrity of mucosal cilia. Ethics committee name: the Animal Laboratory Ethical Committee of the Zhejiang Chinese Medical University; Approval Code: IACUC-20200727-07; Approval Date: 20200727.

#### 4.6.1. H&E Staining

Fixation with formaldehyde, dehydration with a gradient concentration of ethanol, paraffin embedding, xylene dewaxing, dehydration, hematoxylin staining, hydrochloric acid ethanol differentiation, eosin staining, and neutral gum mounting were performed. Changes in tissue morphology were observed in the nasal mucosa of each group of rats (taken on the 8th day).

#### 4.6.2. SEM Observation

After 2.5% glutaraldehyde fixation, PBS buffer cleaning, osmic acid secondary fixation, and gradient concentration ethanol dehydration, electron microscope samples were prepared. SEM was used to observe the morphological changes in nasal mucosa cilia on the 8th, 15th, and 22nd days.

### 4.7. The Effect of AA on Sleep in Insomnia Rats

In the insomnia efficacy test of AA, 42 SD rats (180–220 g, male) were randomly divided into 7 groups with 6 rats in each group, grouped as follows: a. normal group; b. model group (normal saline); c. diazepam group—the diazepam was from Shandong Xinyi Pharmaceutical Co., Ltd. (Shandong, China); d. low-dose group; e. middle-dose group; f. high-dose group; g. a-β-CD group. The SD rats were reared adaptively for one week, with free access to food and water. Except for the normal group, the other groups were intraperitoneally injected with PCPA (4-Chloro-DL-phenylalanine) was supplied by Aladdin company (Shanghai, China) between 9–10 a.m. for 2 days. After screening the dosage of PCPA in the early experiment, the dosage of 450 mg/kg and weak basic normal saline (pH: 7–8) were selected to prepare PCPA into a 1 mL/100 g suspension. From the 3rd day, the model group was given normal saline (50 μL/each) by nasal mucosal administration, and the diazepam group was given a diazepam aqueous solution (0.92 mg/kg, 100 g/mL) by gavage. The low-dose group, medium-dose group, high-dose group, and a-β-CD group were all administered with the respective drugs by nasal mucosal administration (50 μL/each) for 7 days, which was used for the behavioral experiments such as the elevated cross maze. After 7 days, the rats in each group were anesthetized, and the hypothalamic tissues were removed and preserved for subsequent experiments.

### 4.8. Experiments on Pentobarbital-Sodium-Induced Sleep in Rats

After preliminary experiments, the suprathreshold dose of pentobarbital sodium was 45 mg/kg, and the subthreshold dose was 30 mg/kg. After the last administration, each group was intraperitoneally injected with 30 mg/kg pentobarbital sodium to observe and record whether the animals fell asleep. Moreover, each group was intraperitoneally injected with 45 mg/kg pentobarbital sodium to observe the sleep condition of the animals and record the sleep latency and sleep duration [41].

### 4.9. Behavioral Investigation

#### 4.9.1. Self-Activity and Weight Change

The daily body weight and the number of spontaneous activities within 5 min after the last drug administration were recorded.

#### 4.9.2. Open Field Test

After the last administration, the rats were placed in the open field, the animals adapted for 2 min, and the data were collected automatically by Smart 3.0 software. The movement of the rats within 5 min was recorded, and the indexes of experimental rats were recorded, including the movement distance, movement time, stool number, etc., and their movement characteristics were observed.

#### 4.9.3. Tail Suspension Test

After administration, 1/3 of the tail of the rat was fixed with medical tape and hung upside down on the stent, adaptation for 2 min was allowed, the state of the animal was observed, and the immobility time of the rat was recorded within 4 min after the tail was suspended.

#### 4.9.4. Elevated Cross Maze Test

The rats were placed into the experimental environment. The rats were then adapted for 1 min and placed in the central area of the maze with their heads facing any closed arm direction. The data of each index and the activity within 5 min were collected and recorded automatically by Smart 3.0 data software. The observation indexes included the open arm entry times and residence time (two front melons must enter the arm), the closed arm entry times and residence time, and the total entry times in the elevated cross maze.

#### 4.9.5. Detection of Monoamine Neurotransmitters and Amino Acids

After 7 days, the rats were anesthetized, and the hypothalamus tissue of each group was quickly taken. The homogenate was fully homogenized in an ice bath, centrifuged, and analyzed according to the instructions of the ELISA kit. The OD value was detected by a multifunctional enzyme labeling instrument (Synergy H1, Berten Company, Winooski, VT, USA). The expression levels of NE, 5-HT, Glu, and GABA in the hypothalamus of each group of animals were compared, and the ratio of GABA to Glu was determined to be abnormal.

### 4.10. Statistical Analysis

SPSS 20.0 was used for the statistical analysis. Analysis of variance (ANOVA) and the “T” test were used to test the significance the among groups. *p* < 0.05 was considered to be a significant difference, and *p* < 0.01 was considered to be a highly significant difference. During the administration period, each animal only conducted the same behavioral experiment once, and each behavioral experiment was completed.

## Figures and Tables

**Figure 1 molecules-28-00635-f001:**
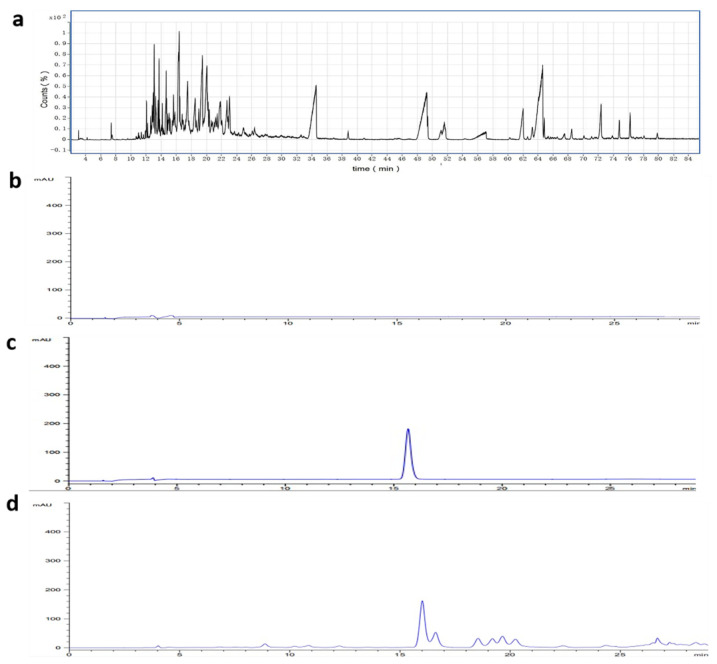
GC–MS total flow diagram of the AEO and HPLC diagram of agarotetraol. (**a**) The GC–MS total flow diagram of AEO; (**b**) HPLC chart of blank control; (**c**) HPLC chart of agarotetraol standard; (**d**) HPLC chart of EA.

**Figure 2 molecules-28-00635-f002:**
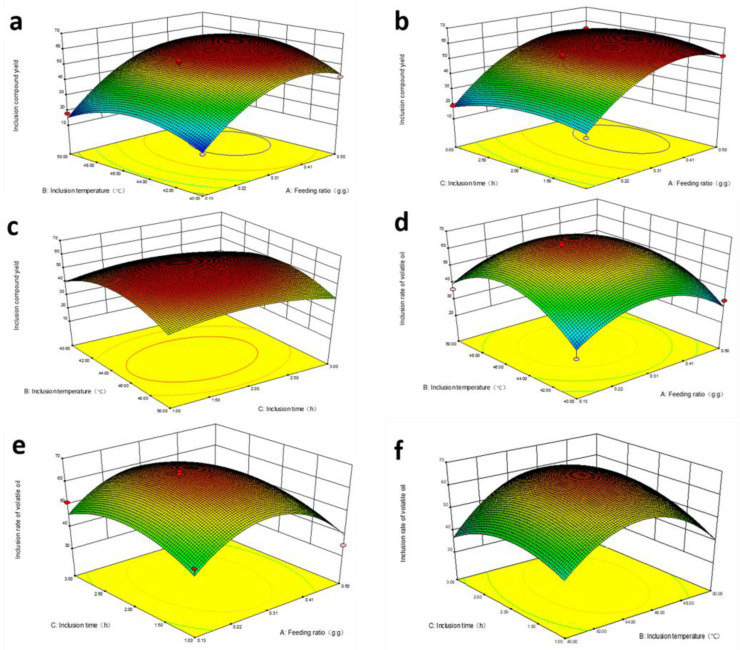
Response surface curve (the 3D plots) showing the relationship between the feeding ratio (g:g), inclusion temperature (°C), and inclusion time (h) toward the response of: (**a**) ICY (%) and (**b**) IEO (%). (**a**) The influence of varying the feeding ratio and inclusion temperature on ICY (%); (**b**) the influence of varying the feeding ratio and inclusion time on ICY (%); (**c**) the influence of varying the inclusion temperature and inclusion time on ICY (%); (**d**) the influence of varying the feeding ratio and inclusion temperature on IEO (%); (**e**) the influence of varying the feeding ratio and inclusion time on IEO (%); (**f**) the influence of varying the inclusion temperature and inclusion time on IEO (%).

**Figure 3 molecules-28-00635-f003:**
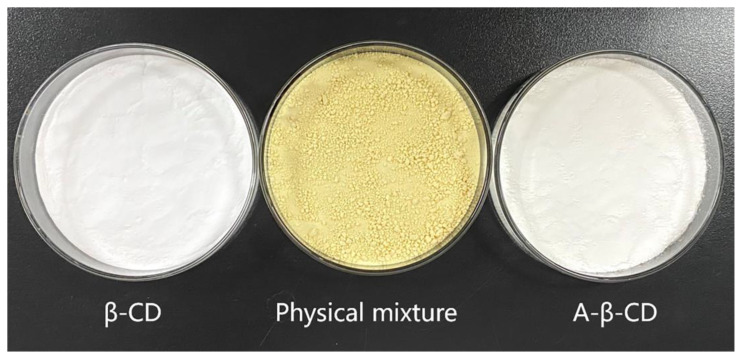
Photographs of β-CD, the physical mixture, and a-β-CD.

**Figure 4 molecules-28-00635-f004:**
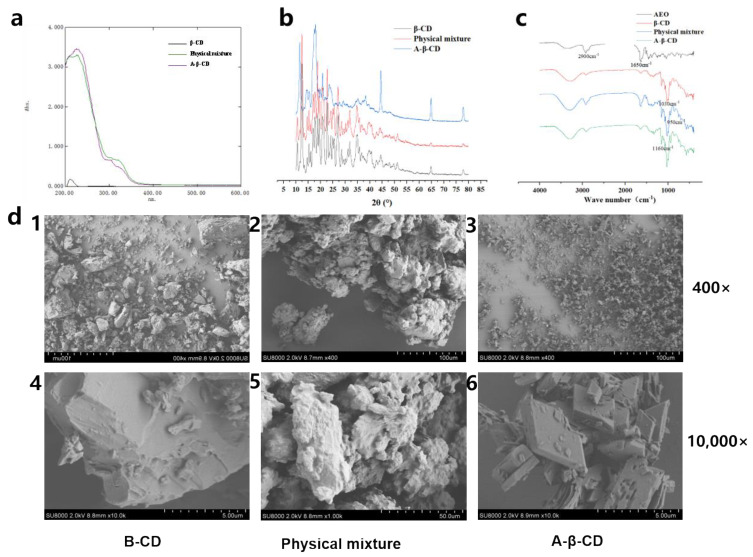
Characterization of a−β−CD. (**a**) UV absorption spectrum; (**b**) FT−IR spectroscopy; (**c**) X−ray powder diffraction; (**d**) scanning electron microscopy: 1. β−CD 400×; 2. physical mixture 400×; 3. A−β−CD 400×; 4. β−CD 10,000×; 5. physical mixture 10,000×; 6. A−β−CD 10,000×.

**Figure 5 molecules-28-00635-f005:**
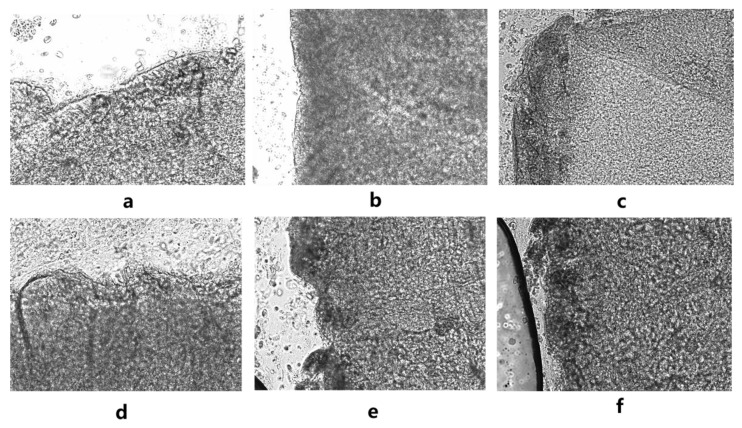
Optical microscope image of the upper jaw mucosa of *Bufo gargarizans* with a magnification of 50×. (**a**) Normal group; (**b**) positive control group; (**c**) low-dose group; (**d**) middle-dose group; (**e**) high-dose group; (**f**) a-β-CD group.

**Figure 6 molecules-28-00635-f006:**
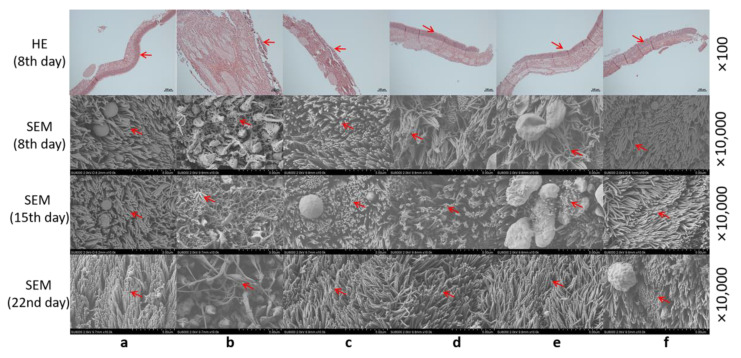
Morphology of the nasal mucosa of the rats in each group. (**a**) Normal group; (**b**) positive control group; (**c**) low-dose group; (**d**) middle-dose group; (**e**) high-dose group; (**f**) a-β-CD group.

**Figure 7 molecules-28-00635-f007:**
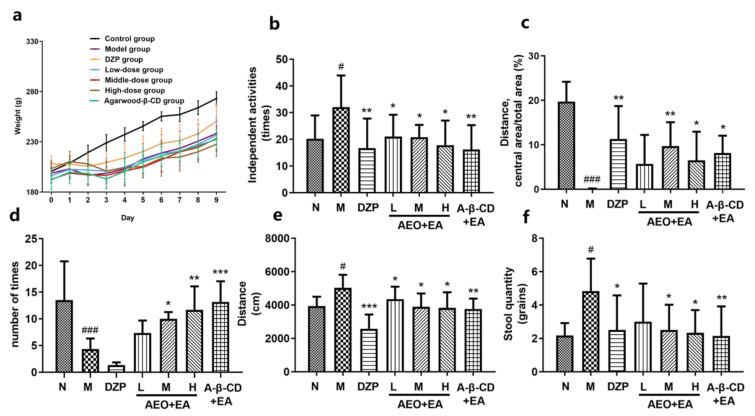
Effects of drugs on the weight change, number of independent activities, and open field test of insomnia animals. (**a**) The weight change, (**b**) the number of independent activities, (**c**) the ratio of the distance between entering the central area and the total area, (**d**) the times of entering the central region, (**e**) the total distance, and (**f**) the number of feces. The low-, middle- and high-dose groups were given the mixture of AEO and EA. The a-β-CD group was given the mixture of a-β-CD and EA. Abbreviations: N—the normal group, M—the model group, DZP—the diazepam group, L—the low-dose group, M—the middle-dose group, H—the high-dose group, and a-β-CD + EA was the a-β-CD group. ^#^ *p* < 0.05 and ^###^ *p* < 0.001, compared to the normal group; * *p* < 0.05, ** *p* < 0.01 and *** *p* < 0.001 compared to the model group. Data were presented as mean ± SD (*n* = 6).

**Figure 8 molecules-28-00635-f008:**
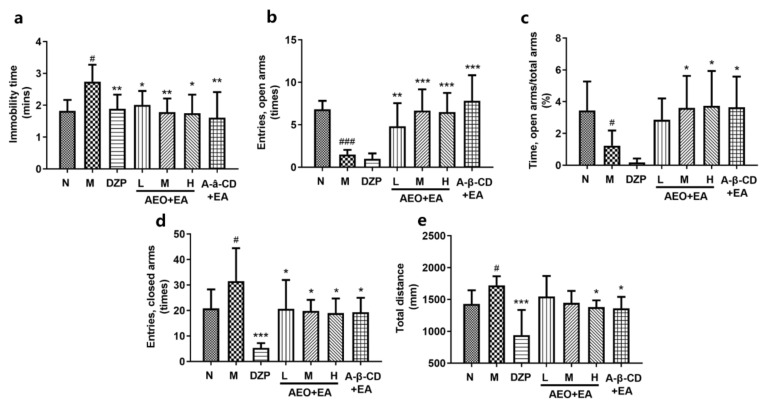
Effects of the tail suspension test and elevated cross maze test. (**a**) The effect on the immobility time in the tail suspension test, (**b**) the number of rats entering the open arms, (**c**) the ratio of the time between entering the open arms and the total arms, (**d**) the number of rats entering the closed arms, (**e**) the total distance into the maze. The level of significance was set at ^#^
*p* < 0.05, ^###^
*p* < 0.001, compared to the normal group; * *p* < 0.05, ** *p* < 0.01 and *** *p* < 0.001 compared to the model group. Data were presented as mean ± SD (*n* = 6).

**Figure 9 molecules-28-00635-f009:**
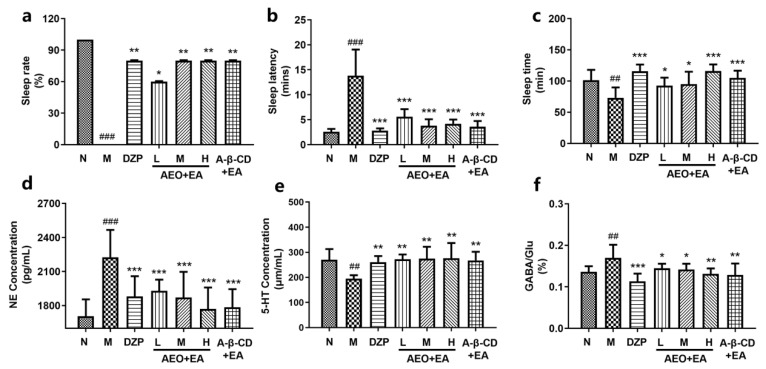
Effects of drugs on the sleep and experimental results of anti-insomnia pharmacodynamics. (**a**) The effect on the sleep rate, (**b**) the effect on sleep latency, (**c**) the effect on sleep time, (**d**) the concentration of NE, (**e**) the concentration of 5-HT, and (**f**) percentage of the ratio of GABA to Glu. The level of significance was set at ^##^ *p* < 0.01 and ^###^ *p* < 0.001, compared to the normal group; * *p* < 0.05, ** *p* < 0.01 and *** *p* < 0.001 compared to the model group. Data were presented as mean ± SD (*n* = 6).

**Table 1 molecules-28-00635-t001:** Chemical composition of the AEO of agarwood.

Sequence	Substance Name	Molecular Formula	RT	Peak Area Ratio (%)
1	Benzaldehyde	C_7_H_6_O	3.18	0.09
2	2-Butanone, 4-phenyl-	C_10_H_12_O	7.42	0.27
3	2-Butanone, 4-(4-methoxyphenyl)-	C_11_H_14_O_2_	11.01	0.23
4	(4S,4aR,6R)-4,4a-Dimethyl-6-(prop-1-en-2-yl)-1,2,3,4,4a,5,6,7-octahydronaphthalene	C_15_H_24_	10.67	0.09
5	α-Santalol	C_15_H_24_O	12.08	0.8
6	2-((2R,4aR,8aS)-4a-Methyl-8-methylenedecahydronaphthalen-2-yl)prop-2-en-1-ol	C_15_H_24_O	12.23	0.26
7	2-((2S,4aR)-4a,8-Dimethyl-1,2,3,4,4a,5,6,7-octahydronaphthalen-2-yl)propan-2-ol	C_15_H_26_O	12.63	0.52
8	Guaiol	C_15_H_26_O	12.87	1.29
9	β-Guaiene	C_15_H_24_	13.02	1.53
10	2-((2R,4aR,8aR)-4a,8-Dimethyl-1,2,3,4,4a,5,6,8a-octahydronaphthalen-2-yl)acrylaldehyde	C_15_H_22_O	13.63	0.65
11	9-Isopropyl-1-methyl-2-methylene-5-oxatricyclo[5.4.0.0(3,8)]undecane	C_15_H_24_O	13.68	0.27
12	2-Naphthalenol, 2,3,4,4a,5,6,7-octahydro-1,4a-dimethyl-7-(2-hydroxy-1-methylethyl)	C_15_H_26_0_2_	13.85	0.47
13	Longifolenaldehyde	C_15_H_24_O	14.22	0.7
14	7-Isopropenyl-1,4a-dimethyl-4,4a,5,6,7,8-hexahydro-3H-naphthalen-2-one	C_15_H_22_O	14.75	3.06
15	Corymbolone	C_15_H_24_O_2_	14.81	0.7
16	Cyclolongifolene oxide, dehydro-	C_15_H_22_O	14.91	0.37
17	(E)-3-((4S,7R,7aR)-3,7-Dimethyl-2,4,5,6,7,7a-hexahydro-1H-inden-4-yl)-2-methylacrylald	C_15_H_22_O	15.14	1.77
18	(1R,4aR,7R,8aR)-7-(2-Hydroxypropan-2-yl)-1,4a-dimethyldecahydronaphthalen-1-ol	C_15_H_28_0_2_	15.25	0.48
19	Methyl-2-((2R,4aR)-4a,8-dimethyl-1,2,3,4,4a,5,6,7-octahydronaphthalen-2-yl)acrylate	C_16_H_24_O_2_	15.53	0.26
20	2H-Cyclopropa[a]naphthalen-2-one1,1a,4,5,6,7,7a,7b-octahydro-1,1,7,7a-tetramethyl-, (1a. α,7. α,7a. α,7b. α)-	C_15_H_22_O	15.71	0.92
21	2(1H)-Naphthalenone,4a,5,6,7,8,8a-hexahydro-6-[1-(hydroxymethyl)ethenyl]-4,8a-dimethyl-, [4ar-(4a. α.,6. α.,8a. β.)]-	C_15_H_22_O_2_	18.68	1.67
22	β-Cyclocostunolide	C_15_H_20_O_2_	20.16	3.31
23	(4aR,5S)-1-Hydroxy-4a,5-dimethyl-3-(propan-2-ylidene)-4,4a,5,6-tetrahydronaphthalen-2(3H)-one	C_15_H_20_O_2_	20.30	3.15
24	(4aS,7R)-7-(2-Hydroxypropan-2-yl)-1,4a-dimethyl-4,4a,5,6,7,8-hexahydronaphthalen-2(3H)-one	C_15_H_24_O_2_	20.40	0.72
25	5,8-Dihydroxy-4a-methyl-4,4a,4b,5,6,7,8,8a,9,10-decahydro-2(3H)-phenanthrenone	C_15_H_22_O_3_	20.53	0.76
26	Eudesma-5,11(13)-dien-8,12-olide	C_15_H_20_O_2_	20.64	0.53
27	6-(1-Hydroxymethylvinyl)-4,8a-dimethyl-3,5,6,7,8,8a-hexahydro-1H-naphthalen-2-one	C_15_H_20_O_2_	22.12	1.39
28	2aS,3aR,5aS,9bR)-2a,5a,9-Trimethyl-2a,4,5,5a,6,7,8,9b-octahydro-2H-naphtho[1,2-b]oxireno[2,3-c]furan	C_15_H_22_O_2_	22.96	1.9
29	Propanoic acid, 2-methyl-, (dodecahydro-6a-hydroxy-9a-methyl-3-methylene-2,9-dioxoazuleno [4,5-b] furan-6-yl)methyl ester, [3aS-(3a. α.,6.β.,6a. α.,9a.β.,9b. α.)] -	C_19_H_26_O_6_	25.19	0.55
30	1,5-diphenyl-1-Penten-3-one	C_17_H_16_O	26.63	0.5
31	10-epi-γ--Eudesmol	C_15_H_26_O	27.47	0.08
32	Agarospirol	C_15_H_26_O	29.23	0.93
33	2-Phenethyl-4H-chromen-4-one	C_17_H_14_O_2_	34.89	1.56
34	γ--Gurjunenepoxide-(2)	C_15_H_24_O	43.09	1.15
35	2-(4-Methoxyphenethyl)-4H-chromen-4-one	C_18_H_16_O_3_	49.82	0.75
36	Acetamide, N-(4-benzyloxyphenyl)-2-cyano-	C_16_H_14_N_2_O_2_	55.59	0.26
37	13-Docosenamide, (Z)-	C_22_H_43_NO	60.28	0.09
38	Squalene	C_30_H_50_	62.28	0.63
39	Methoxy-2-(4-methoxyphenethyl)-4H-chromen-4-7-one	C_19_H_18_O_4_	63.38	0.77
40	6-Methoxy-2-phenethyl-4H-chromen-4-one	C_18_H_16_O_3_	63.60	0.13
41	6,7-Dimethoxy-2-phenethyl-4H-chromen-4-one	C_19_H_18_O_4_	64.93	9.67
42	4H-1-Benzopyran-4-one, 5-hydroxy-7-methoxy-2-(4-methoxyphenyl)-	C_17_H_14_O_5_	65.86	0.4
43	6,7-Dimethoxy-2-(4-methoxyphenethyl)-4H-chromen-4-one	C_20_H_20_O_5_	72.54	2
44	Stigmasterol	C_29_H_48_O	74.88	0.55
45	γ—Sitosterol	C_29_H_50_O	76.33	0.74
46	Spinasterone	C_29_H_46_O	79.53	4.4

**Table 2 molecules-28-00635-t002:** The results of the single factor investigation into the inclusion process of β-CD.

Factors		ICY (%)	OIC (%)	IEO (%)
Feeding ratio (g:g)	0.13:1.40	30.17 ± 0.02	8.50 ± 0.12	31.53 ± 0.71
	0.25:1.40	53.03 ± 1.63 ^##^	17.37 ± 0.70 ^##^	59.67 ± 3.09 ^##^
	0.50:1.40	63.21 ± 0.33 ^##,^**	21.82 ± 1.12 ^##,^**	52.35 ± 2.69 ^##^
Inclusion temperature (°C)	40.00	52.76 ± 1.61	16.92 ± 0.46	58.79 ± 2.10
	45.00	53.03 ± 1.63	17.37 ± 0.70	59.67 ± 3.09
	50.00	48.77 ± 1.80	16.66 ± 0.83	53.73 ± 4.26
Inclusion time (h)	1.00	49.46 ± 3.07	14.31 ± 0.40	46.68 ± 3.87
	2.00	53.03 ± 1.63	17.37 ± 0.70	59.67 ± 3.09
	3.00	56.81 ± 0.42	17.06 ± 0.42	63.04 ± 1.54

Values represent the mean ± SD of three the independent measurements. Three factors were selected as the main investigation aspects: (a) feeding ratio (the ratio of essential oil to β-CD, g:g): 0.13:1.40, 0.25:1.40, and 0.50:1.40; (b) inclusion temperature: 40 °C, 45 °C, and 50 °C; (c) inclusion time: 1 h, 2 h, and 3 h. ICY (%) = the inclusion compound yield, OIC (%) = oil content of the inclusion compound, and IEO (%) = inclusion rate of the essential oil. ICY (%), OIC (%), and IEO (%) were taken as the indexes to determine the optimal values as the basis for the subsequent design of the experiment. The level of significance was set at ^##^
*p* < 0.01, compared to the 0.13:1.40 group; ** *p* < 0.01 compared to the 0.25:1.40 group.

**Table 3 molecules-28-00635-t003:** Experimental design and results.

Run	A. Feeding Ratio	B. Inclusion Temperature	C. Inclusion Time	ICY (%)	IEO (%)
1	0.13	40	2	15.92	27.01
2	0.25	50	3	29.39	45.69
3	0.25	45	2	43.04	67.59
4	0.5	45	1	52.11	34.90
5	0.5	50	2	51.02	36.60
6	0.25	45	2	55.99	61.60
7	0.25	45	2	55.24	65.29
8	0.25	40	3	31.53	35.46
9	0.25	45	2	54.53	64.50
10	0.25	45	2	50.36	68.65
11	0.5	40	2	42.28	33.36
12	0.13	45	1	21.66	46.70
13	0.13	50	2	17.85	36.29
14	0.5	45	3	54.86	36.29
15	0.25	40	1	37.36	41.74
16	0.13	45	3	18.88	51.13
17	0.25	50	1	36.14	39.96

**Table 4 molecules-28-00635-t004:** Analysis of variance of the ICY% and IEO% index model.

Source	ICY%					IEO%				
	Squares	df	Square	Value	Prob > F	Squares	df	Square	Value	Prob > F
Model	3208.92	9	356.55	16.46	0.0006 **	2862.54	9	318.06	12.37	0.0016 **
A—Feeding ratio (g:g)	1983.11	1	1983.11	91.56	<0.0001 **	49.87	1	49.87	1.94	0.2064
B—Inclusion temperature (°C)	12.39	1	12.39	0.57	0.4742	44.14	1	44.14	1.72	0.2315
C—Inclusion time (h)	12.34	1	12.34	0.57	0.475	2.57	1	2.57	0.10	0.7610
AB	19.85	1	19.85	0.92	0.3703	6.81	1	6.81	0.26	0.6227
AC	13.78	1	13.78	0.64	0.4513	0.93	1	0.93	0.036	0.8546
BC	0.21	1	0.21	9.70 × 10^−3^	0.9243	36.02	1	36.02	1.40	0.2753
A^2	740.73	1	740.73	34.20	0.0006 **	836.52	1	836.52	32.52	0.0007 **
B^2	573.13	1	573.13	26.46	0.0013 **	1198.82	1	1198.82	46.61	0.0002 **
C^2	181.13	1	181.13	8.36	0.0233 *	265.15	1	265.15	10.31	0.0148 *
Residual	151.61	7	21.66			180.06	7	25.72		
Lack of Fit	35.94	3	11.98	0.41	0.7525	149.57	3	49.86	6.54	0.0506
Pure Error	115.67	4	28.92			30.49	4	7.62		

* *p* < 0.05, ** *p* < 0.01.

**Table 5 molecules-28-00635-t005:** Effect of AA on Bufo toad maxillary mucosa.

	In Vitro		In Vivo	
Group	PVD (min)	PVD %	PVD (min)	PVD%
Normal group	1651 ± 82.92 ***	100 ***	1669 ± 79.23 ***	100 ***
Positive control group	0 ± 0 ^###^	0 ^###^	0 ± 0 ^###^	0 ^###^
Low-dose group	1608 ± 58.78 ***	97.4 ***	1617 ± 73.08 ***	96.88 ***
Middle-dose group	1607 ± 74.20 ***	97.33 ***	1609 ± 136.78 ***	96.38 ***
High-dose group	1583 ± 113.93 ***	95.88 ***	1598 ± 160.99 ***	95.74 ***
A-β-CD	1610 ± 72.72 ***	97.52 ***	1625 ± 75.37 ***	97.34 ***

The level of significance was set at ^###^
*p* < 0.001, compared to the normal group; *** *p* < 0.001 compared to the model group. Data were presented as mean ± SD (*n* = 6).

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
