# Peer review of "The Effect of Chinese Agarwood Essential Oil with Cyclodextrin Inclusion against PCPA-Induced Insomnia Rats"

_molecules, 2023, doi:10.3390/molecules28020635_

Round 1

Reviewer 1 Report

The manuscript "Study on the Effect of the Active Components of Chinese Agarwood and Its Cyclodextrin Inclusion Compound via the Nasal Mucosa Administration Route on the Sleep of PCPA-Induced Insomnia Rats "  presents the results of their studies on the safety and effectiveness of the extracted agarwood active ingredients (AA) in the treatment of insomnia rats by nasal administration. The idea seems to be original and the experiments were carried out correctly. However, the manuscript has shortcomings that prevent its publication in its current form.

The title "Study on the Effect of the Active Components of Chinese Agarwood and Its Cyclodextrin Inclusion Compound via the Nasal Mucosa Administration Route on the Sleep of PCPA-Induced Insomnia Rats" is too long and inappropriate. 

The quality of figures 1,4,5 and 6 is low and must be improved.

Please add a space between the words and citation numbers in all places in the manuscript "inclusion compound[21,33,34]."

The first column in Table 1 needs to be revised and corrected.

In Table 2, the no. of significant figures of all numbers must be uniform.

Lines 32-35: The equation has to be clarified and written in a proper way using the equation function.

The findings of the characterization tests must be discussed more and supported with references.

Author Response

1. The title "Study on the Effect of the Active Components of Chinese Agarwood and Its Cyclodextrin Inclusion Compound via the Nasal Mucosa Administration Route on the Sleep of PCPA-Induced Insomnia Rats" is too long and inappropriate. 

Response 1: Thanks for your thoughtful comments and suggestion. We change the title into “Effect Chinese Agarwood Essential Oil with Cyclodextrin Inclusion Against PCPA-Induced Insomnia Rats”.

2. The quality of figures 1,4,5 and 6 is low and must be improved.

Response 2: As recommended, we have revised the figures 1, 4, 5 and 6.

3. Please add a space between the words and citation numbers in all places in the manuscript "inclusion compound [21,33,34]."

Response 3: Thank you. As recommended, we have added all spaces between the words and citation in this manuscript.

4. The first column in Table 1 needs to be revised and corrected.

Response 4: Thank you for your valuable advice. We have revised and corrected the Table 1.

5. In Table 2, the no. of significant figures of all numbers must be uniform.

Response 5: Thank you very much for your suggestion. As recommended, we have revised the Table 2 and manuscript.

6. Lines 32-35: The equation has to be clarified and written in a proper way using the equation function.

Response 6: Thank you for your valuable advice. The equation has to be modified in a proper way using the equation function.

7. The findings of the characterization tests must be discussed more and supported with references.

Response 7: As suggested, we have added more discussions and supporting references in "2.2.5-2.2.7".

Reviewer 2 Report

It is interesting to read the manuscript “Study on the Effect of the Active Components of Chinese Agarwood and Its Cyclodextrin Inclusion Compound via the Nasal Mucosa Administration Route on the Sleep of PCPA-Induced Insomnia Rats”. I appreciate the authors' efforts in conducting the study, which will be more beneficial to scholars working in the field. However, if the following changes are made and they are incorporated into the manuscript, it could be considered for publication in Molecules Journal.

 1.    Because the title is too long and a little confusing, I suggest authors to change it. I suggest “Effect Chinese Agarwood Essential Oil with Cyclodextrin Inclusion Against PCPA-Induced Insomnia Rats”. If necessary, the authors can modify the suggested title.

2.    Make the following sentence in the abstract better. “Three factors have significant effects on ICY% and IEO% of a-β-CD, and the order of influence was: A. inclusion ratio > B. inclusion temperature > C. inclusion time”. For example “Inclusion ratio, temperature and time are the three factors have significant effects on ICY% and IEO% of a-β-CD”.

3.    The problem statement for this research in the introduction does not support the authors' overall research (Line 56-62).

4.    α-santalol, β-Guaiene, and γ-Sitosterol. All through the manuscript, pay attention to the symbol. Since some locations have full names while others have symbols.

5.    Results and Discussion will be used as the new title for Section 2. There is great writing in the results section. The lack of explanation in the discussion of each segment was another thing I noticed. Instead of giving more background information about the literature, I would advise the authors to refocus their discussion on each part so that it is evident how the research's findings fit into the larger context of what is happening right now in agarwood essential oil and insomnia.

6.    The author gave a general description of the completed research in the conclusion. I suggest that authors to offer a critical justification on their findings and observations. As a result, everyone will understand the value of this research. Potential points of view must be covered in the conclusion as well. The importance of this study piece should be emphasised by the author.

7.    Table 5, Where are the values for the model group?

8.    Include magnification for Figures 5 and 6.

9.    Information regarding the type of rats used, how they are maintained for, and further details like the reference number and information on the ethical committee's approval are not provided. This information must be provided.

10.  min or minutes, keep it consistent throughout the manuscript.

11.  Line 444-445 to be moved to materials section (4.1)

Author Response

Point 1: Because the title is too long and a little confusing, I suggest authors to change it. I suggest “Effect Chinese Agarwood Essential Oil with Cyclodextrin Inclusion Against PCPA-Induced Insomnia Rats”. If necessary, the authors can modify the suggested title.

Response 1: Thank you for your valuable advice. We modify the title as “Effect Chinese Agarwood Essential Oil with Cyclodextrin Inclusion Against PCPA-Induced Insomnia Rats”

Point 2: Make the following sentence in the abstract better. “Three factors have significant effects on ICY% and IEO% of a-β-CD, and the order of influence was: A. inclusion ratio > B. inclusion temperature > C. inclusion time”. For example “Inclusion ratio, temperature and time are the three factors have significant effects on ICY% and IEO% of a-β-CD”.

Response 2: Thank you. We modify the abstract in this manuscript.

Point 3: The problem statement for this research in the introduction does not support the authors' overall research (Line 56-62).

Response 3: Thank you for your valuable advice. We have modified the “introduction” in the line 71-74.

Point 4: α-santalol, β-Guaiene, and γ-Sitosterol. All through the manuscript, pay attention to the symbol. Since some locations have full names while others have symbols.

Response 4: Thanks for your suggestion. We have checked this problem through the manuscript and revised.

Point 5: Results and Discussion will be used as the new title for Section 2. There is great writing in the results section. The lack of explanation in the discussion of each segment was another thing I noticed. Instead of giving more background information about the literature, I would advise the authors to refocus their discussion on each part so that it is evident how the research's findings fit into the larger context of what is happening right now in agarwood essential oil and insomnia.

Response 5: Thank you for your valuable advice. We have revised the title for Section2 and added discussion to the manuscript.

Point 6: The author gave a general description of the completed research in the conclusion. I suggest that authors to offer a critical justification on their findings and observations. As a result, everyone will understand the value of this research. Potential points of view must be covered in the conclusion as well. The importance of this study piece should be emphasised by the author.

Response 6: Thanks for your thoughtful comments and suggestion. For your suggestion, we revised it in the “Conclusion”.

Point 7: Table 5, Where are the values for the model group?

Response 7: Through Bufo gargarizans mucosa cilia test and SD rat in vivo test, observed Bufo gargarizans mucosa and rat nasal mucosa cilia after being destroyed. In this study, 1% sodium deoxycholate was used as a positive control group to study the effect of AA on mucosal integrity and cilia, and to evaluate its nasal toxicity comprehensively.

Point 8: Include magnification for Figures 5 and 6.

Response 8: Thank for your suggestion. we have added the magnification in the Figures 5 and 6.

Point 9: Information regarding the type of rats used, how they are maintained for, and further details like the reference number and information on the ethical committee's approval are not provided. This information must be provided.

Response 9: Thank you for bringing our attention to this neglect. The supplementary information is as follows: Ethic Committee Name: the Animal Laboratory Ethical Committee of the Zhejiang Chinese Medical University; Approval Code: IACUC-20200727-07; Approval Date: 20200727. At the same time, we add the information in line 380-382.

Point 10: min or minutes, keep it consistent throughout the manuscript.

Response 10: As recommended, we have carefully modified and keep it consistent throughout the manuscript.

Point 11: Line 444-445 to be moved to materials section (4.1)

Response 11: Thank you. Line 444-445 are moved to materials section (4.1).

Reviewer 3 Report

In the manuscript, ‘Study on the Effect of the Active Components of Chinese Agarwood and Its Cyclodextrin Inclusion Compound via the Nasal Mucosa Administration Route on the Sleep of  PCPA-Induced Insomnia Rats’, the authors have characterized agarwood essential oil (AEO) and a-b-CD prepared from AEO. The authors found that nasal administration of agarwood active ingredients (AA) was safe and had therapeutic benefits on rats modelled for sleep disorder. The findings in the study are interesting and the manuscript can be improved by addressing the following concerns:

·       Table 2: Did the authors perform a statistical test to establish significance between the varying feeding ratios? This should be mentioned in the figure legend.

·       Description of Section 2.2.3 is confusing and should be simplified for easier understanding.

·       What are the standard tests performed apart from FT-IR, SEM and XRD to demonstrate success of a-b-CD preparation.

·       Section 2.3.1: The authors should explain the rationale of the test.

·       Figure 6: The authors should use arrows to clearly show the structures of interest.

·       Figure 6: What are the rounded structures observed for ‘High dose group’ in the SEM images for Day 8 and Day 15?

Author Response

Point 1: Table 2: Did the authors perform a statistical test to establish significance between the varying feeding ratios? This should be mentioned in the figure legend.

Response 1: Thanks for your thoughtful comments and suggestion. We have performed the statistical test to establish significance between the varying feeding ratios and revised in the Table 2.

Point 2: Description of Section 2.2.3 is confusing and should be simplified for easier understanding.

Response2: Thank for your suggestion. We have simplified the “Section 2.2.3” in line 85-87.

Point 3: What are the standard tests performed apart from FT-IR, SEM and XRD to demonstrate success of a-b-CD preparation.

Response3: Firstly, we judged that a-β-CD was not a physical mixture by its properties. Then, we used UV-Vis Spectrophotometer to scan the β-CD, physical mixture and a-β-CD inclusion complex at full wavelength, the maximum absorption wavelength of a-β-CD inclusion complex at 227 nm is different from that of blank cyclodextrin and physical mixture. Futher, FT-IR, SEM and XRD were used to characterize the formation of a-β-CD inclusion complex.

Point 4: Section 2.3.1: The authors should explain the rationale of the test.

Response 4: The evaluation methods of ciliary toxicity of nasal mucosa are usually in vitro, in vivo and in vivo. The common animal models are toad maxillary mucocilia, frog maxillary mucocilia, chick embryo trachea mucocilia, etc. It is a simple and reliable experimental method for Bufo gargarizans maxillary mucosa because of its convenience, obvious observation and large mucosa area. Therefore, we used it in vitro and in vivo experiments of toad maxillary cilia to simulate the safety of nasal mucosa.

Point 5: Figure 6: The authors should use arrows to clearly show the structures of interest.

Response 5: Thank you for your valuable advice. We modified Figure 6 to mark the structures of interest with arrows.

Point 6: Figure 6: What are the rounded structures observed for ‘High dose group’ in the SEM images for Day 8 and Day 15?

Response 6.: In the figure6, the nuclei of olfactory neurons and basal cells were condensed, vacuoles were formed, and necrosed and shed in the second week. The remaining basal cells began to transform into bulbous basal cells, and the olfactory epithelium recovered in the fourth week. Therefore, the rounded structures observed for ‘High dose group’ in the SEM images for Day 8 and Day 15 are vacuoles.

Round 2

Reviewer 1 Report

The article can be accepted in the present form.

Reviewer 2 Report

I really appreciate the authors' effort because they made significant changes to the manuscript in response to my comments. As a result, I recommend considering it for publication in the Molecules Journal after the minor modification in the title.

The authors can choose anyone of the following title

Chinese Agarwood Essential Oil with Cyclodextrin Inclusion Against PCPA-Induced Insomnia Rats

 Or

Effect of Chinese Agarwood Essential Oil with Cyclodextrin Inclusion Against PCPA-Induced Insomnia Rats